# KOKOYI: EXECUTABLE LATEX FOR END-TO-END DEEP LEARNING

## ABSTRACT

Despite substantial efforts from the deep learning system community to relieve researchers and practitioners from the burden of implementing models with ever-growing complexity, a considerable lingual gap remains between developing models in the language of mathematics and implementing them in the languages of computer. The mission of KOKOYI is to close this gap by enabling automatic translation of mathematics into efficient implementations, thereby making math-in-codes and math-in-model consistent. This paper presents our first step towards the goal: `kokoyi-lang`, a programming language with the syntax of LATEX and the semantics of deep learning mathematics, and a prototype `kokoyi-lang` compiler and runtime supporting advanced optimizations such as auto-batching. KOKOYI is integrated with Jupyter Notebook, and will be released in open-source.

## 1 INTRODUCTION

The success of deep learning is a tale of two ends. At one end is model development, which leverages the language of mathematics to define models. At the other end is model implementation, which relies on programming languages such as Python and CUDA to unleash the power of big data and efficient computation. Despite the success, the language gap between the two ends is arguably the source of many evils, such as slow prototyping, wrong/inefficient/unreadable implementations, irreproducible results in publications, and learning barriers to name but a few. A model is often implemented and re-implemented in different, incompatible deep learning frameworks, an unnecessary fragmentation from a model point of view.

The mission of KOKOYI is to close the language gap between the two ends, making math-in-codes and math-in-model consistent. KOKOYI advocates a simple principle: *your model is your code*. KOKOYI introduces a programming language called `kokoyi-lang` with a programming model elevated to mathematical equations written in LATEX, a rendering language popular among deep learning researchers (Lamport, 1994). Syntactically, `kokoyi-lang` is a LATEX subset that is prevalent in mathematical definitions of deep learning models, making `kokoyi-lang` programs renderable just like LATEX documents. Semantically, `kokoyi-lang` formalizes the consensus of the semantics of mathematics among deep learning researchers and practitioners. KOKOYI aims at boosting productivity at a community level: researchers code a model once instead of twice (one in a program and another in the publication), `kokoyi-lang` is designed to be framework agnostic, the model written in `kokoyi-lang` in a paper is executable by all and can be ported to different frameworks.

As a teaser, the following (rendered) line of `kokoyi-lang` code defines the mean squared error loss function $\ell$ for a linear regression model with weight $w \in \mathbb{R}^d$ and bias $b \in \mathbb{R}$ given a dataset $D$:

$$\ell(w, b, D) \leftarrow \frac{1}{|D|} \sum_{(x,y) \in D} ((w \cdot x + b) - y)^2$$

Given `kokoyi-lang` programs, KOKOYI can generate executables using various backends. In its current prototype, KOKOYI generates PyTorch code because of its widespread adoption. However, compared to PyTorch's deep learning frameworks that rely on Python for usability, KOKOYI can potentially avoid Python's overheads without degrading user experience by translating `kokoyi-lang` programs into more efficient deep learning IRs such as Relay (Roesch et al., 2018) instead of Python.

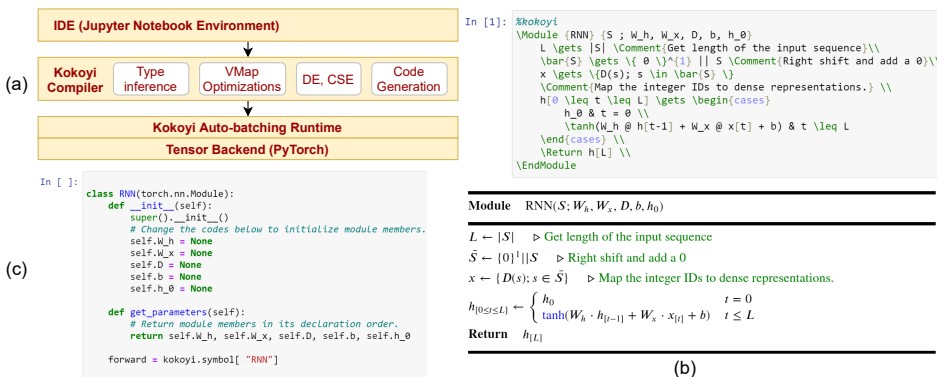

Figure 1: (a) A typical work flow when using KOKOYI; (b) Jupyter notebook example of a multi-layer RNN encoder; (c) The corresponding auto-generated module in PyTorch.

The level of abstraction offered by the language of mathematics can enable various code optimizations deemed hard for existing deep learning frameworks. An example is auto-batching, which can relieve users from the burden of manually vectorizing per-sample code, an often counter-intuitive endeavor mandated solely by efficiency. Previous attempts of auto-batching like TensorFlow-Fold (Looks et al., 2017), DyNet (Neubig et al., 2017) and JAX (Bradbury et al., 2018) heavily rely on Just-in-time compilation, which are limited to statically shaped samples. By contrast, KOKOYI takes a holistic design, constructs built during compilation facilitate auto-batching during execution. As a result, KOKOYI can auto-batch complex models, such as the auto-regressive attention module in Transformer (Vaswani et al., 2017).

KOKOYI focuses on model specification. Our informal survey of sampled papers from several machine learning conferences, showing KOKOYI can impact about 15% to 66% of the work where model development is the focus (Appendix A). As such, we do not attempt to reinvent all the wheels, compiled `kokoyi-lang` programs are callable Python objects and integrate with other stages (e.g. data preparation, model tuning, and model serving) of the deep learning pipeline; whenever possible, KOKOYI reuses efficient existing modules.

We implemented a Jupyter Notebook frontend to facilitate interactive rendering and execution of `kokoyi-lang` programs ((see HTML examples in the supplementary materials). There is also an interactive Web server where one can prototype a model and download both the PyTorch module and the LATEX snippet. We envision supporting compilation of `kokoyi-lang` code embedded in LATEX documents in the future to enable *executable research papers*.

We implemented in `kokoyi-lang` several important deep learning models with various complexity, and find most of our per-sample implementations, with PyTorch as backend, achieve performance comparable to manually tensorized PyTorch implementations. At the same time, models implemented in `kokoyi-lang` are substantially more succinct, for instance the standard Transformer (Vaswani et al., 2017) needs only 65 lines of code, saving users from the tedious and often error-prone process of manually batching computation (Appendix B). Although a large-scale user experience study is yet to be conducted, initial feedbacks from a group of deep learning researchers and data scientists indicate that compared to existing deep learning frameworks, `kokoyi-lang` is much closer to their mathematical formulations of deep learning models. KOKOYI will open source soon.

## 2 KOKOYI OVERVIEW

KOKOYI aims to be pragmatic, while adhering to the "your model is your code" principle. It is built around and leverages the existing Python ecosystem, yet still reserves the flexibility to revert to more efficient execution when possible. Fig. 1(a) depicts the work flow when using KOKOYI. Development starts in an IDE and we currently choose the Jupyter Notebook, a popular interactive environment for live coding and data rendering.

Users can write a KOKOYI program – one or more mathematical equations – in a Jupyter Notebook cell, led by the `%kokoyi` marker to register a hook function that renders the equation via MathJax.

Programming in KOKOYI uses `kokoyi-lang`, a LaTeX dialect that is extended with syntax familiar to DL researchers. Running the cell triggers the compilation of the equation to generate a callable python object. In this example (Fig. 1(b)), the returned object represents the module definition of an $L$-layer RNN to compute the representation of a sentence $S$. We hook the program execution with a proper error handler to highlight the portion of the equations that causes the error.

KOKOYI translates tensors and functions defined in `kokoyi-lang` into tensors in underlying DL frameworks and Python functions, and makes them accessible in a Python dictionary (named `kokoyi.symbols` in Jupyter notebooks). For example, the RNN PyTorch module calls `kokoyi.symbols["RNN"]` (Fig. 1(c)) in its forward function. KOKOYI also ports many PyTorch's commonly used modules (e.g. `\ReLu`); the porting is mostly straightforward, with the only complexities arise from supporting auto-batching. Our current prototype enables one-click auto-generation of a Pytorch module (Fig. 1(c)) once it is specified in `kokoyi-lang`. This capacity is simplistic for now and the user is required to complete the rest of the initialization (e.g. selecting the initializer). It is relatively straightforward to generate more initialization code in the future.

The KOKOYI compiler generates PyTorch code directly and thus the execution is just like any other PyTorch programs. Note that KOKOYI can in principle completely by-pass Python by translating the program into low-level IRs such as Relay IR (Roesch et al., 2018) and LLVM IR (Lattner & Adve, 2004) to enable efficient execution. Nevertheless, we choose to focus on Python code generation mainly due to its dominating ecosystem within the deep learning community. For example, a user can easily replace the model definition in Python with KOKOYI-generated codes, while still utilizing the rich code base from the Python community for data preparation, model tuning, serving and visualization, as all the examples in the supplementary do. KOKOYI itself leverages compilers specialized for Python deep learning program such as TorchScript (PyTorch, 2021) to further accelerate the generated code.

The design of KOKOYI compiler is covered in Section 4. KOKOYI performs auto-batching, where the model is written out entirely from a single sample point of view, but processes *batches* of them after compilation (see Section 5).

## 3 PROGRAMMING IN KOKOYI

Our goal is to design KOKOYI in such a way that it requires almost no learning for researchers who have written papers. In a sense, `kokoyi-lang` is a dialect of LaTeX. KOKOYI can typeset mathematics written in `kokoyi-lang` as if they were written in LaTeX. All mathematics in this section are typeset, parsed, and compiled by KOKOYI, thereby removing the gap between math-in-paper and math-in-code. `kokoyi-lang` is also more than a LaTeX dialect. Mathematics written in `kokoyi-lang` are executable programs carrying the semantics that most members of the DL community will reasonably expect when reading them.

The immediate challenge we face is that math formulations are immensely flexible. Repurposing LaTeX as the programming interface must therefore allow practical and efficient code generation without losing expressiveness. In contrast to programming languages that require alphanumeric variable names, `kokoyi-lang` allows programmers to denote variables by math symbols typeset in LaTeX. The "real estate" of the sub- and superscription needs particular attention. After several iterations, we decide to leave them as part of variable name and impose reasonable constraints elsewhere. For instance, we choose $f(x; \theta)$ instead of $f_\theta(x)$ for learnable functions (i.e. modules), and `x[i]` (typeset as $x_{[i]}$) for indexing. We also introduce common Python syntax where there is no ambiguity such as raising a power and various forms of multiplication, see supplementary for concrete examples.

We now proceed to introduce `kokoyi-lang`'s key constructs: tensors, functions and modules.

**Tensor definition**  A key construct in `kokoyi-lang` is *tensor definition*, which enables one to define ($\leftarrow$) a tensor as the value of an expression, with the command `\gets`:

$$[\textit{tensor}] \leftarrow [\textit{expression}]$$

Expressions in tensor definitions can be applications of built-in tensor operations, such as transpose ($^\top$), element-wise arithmetic (including element-wise reductions such as sum $\sum$ and product $\prod$), dot product ($\cdot$), and indexing ($\cdot_{[\cdot]}$).

The expression in a tensor definition can also be a tensor constructor, which gives an entry-by-entry definition of a tensor. For example, one can define the $M \times N$ attention matrix in Transformer (Vaswani et al., 2017) as

$$\left\{ \left\{ \text{softmax} \left( \frac{q_{[i]}^\top \cdot k_{[j]}}{\sqrt{d}} \right) \right\}_{j=0}^N \right\}_{i=0}^M \qquad \left\{ \left\{ \text{softmax} \left( \frac{q_{[i]}^\top \cdot k_{[j]}}{\sqrt{d}} \right) \right\}_{j=0}^i \right\}_{i=0}^M \qquad (1)$$

where $d$ is embedding dimension, and $q$ and $k$ are $M \times d$ and $N \times d$ matrices consisting of query and key vectors, respectively. The one on the left is full self-attention whereas the right is auto-regressive, i.e. only attentive to tokens behind the current one; in a sequence-to-sequence model, the first is often used in encoder and the second in decoder. One can similarly define a multi-head attention matrix, by adding another index that ranges over $[0, H)$, where $H$ is the number of heads (see supplementary for the full Transformer model).

One can also define a tensor in `kokoyi-lang` by giving a recurrence relation among its parts, such as rows and columns. For example, the hidden state matrix of a vanilla recurrent neural network (RNN) can be given by

$$h_{[0 \le t \le L]} \leftarrow \begin{cases} h_0 & t = 0 \\ \tanh(W_h \cdot h_{[t-1]} + W_x \cdot x_{[t]} + b) & otherwise \end{cases} \qquad (2)$$

where $x$ is a matrix with $L$ rows, each being the embedding of an input token, and $W_h$, $W_x$, and $h_0$ are learnable parameters. This is an example of KOKOYI module that we will describe next. To facilitate compiler optimization, programmers need to annotate indices where recurrence relations exist with a range (e.g. $h_{[0 \le t \le L]}$), which can depend on the dimensions of other tensors as in the case of tensor constructor. Also notice the use of `\begin{cases}` and `\end{cases}` (rendered as the big left brace {) in Eq. (2) for branching (Fig. 1(b))

It is common for a recurrence relation to involve more than one tensor. For example, the hidden state and memory cell of a long short-term memory (LSTM) network (Hochreiter & Schmidhuber, 1997) depends on the network's forget, input, and output gates, which in turn depends on the previous hidden state and memory cell. This recursive mutual dependency can be expressed in `kokoyi-lang` as

$$f_{[1 \le t \le L]} \leftarrow \sigma(U_f \cdot h + V_f \cdot x_{[t]} + b_f) \qquad\qquad i_{[1 \le t \le L]} \leftarrow \sigma(U_i \cdot h + V_i \cdot x_{[t]} + b_i)$$

$$o_{[1 \le t \le L]} \leftarrow \sigma(U_o \cdot h + V_o \cdot x_{[t]} + b_o) \qquad\qquad \tilde{c}_{[1 \le t \le L]} \leftarrow \sigma(U_c \cdot h + V_c \cdot x_{[t]} + b_c)$$

$$c_{[1 \le t \le L]} \leftarrow \begin{cases} c_0 & t = 0 \\ f_{[t]} \circ c_{[t-1]} + i_{[t]} \circ \tilde{c}_{[t]} & otherwise \end{cases} \qquad h_{[1 \le t \le L]} \leftarrow \begin{cases} h_0 & t = 0 \\ o_{[t]} \circ \tanh(c_{[t]}) & otherwise \end{cases}$$

**Function and module** The syntax for tensor definition applies to function definitions as well. The following is a definition of the sigmoid function:

$$\sigma(x) \leftarrow \frac{1}{1 + e^{-x}}$$

`kokoyi-lang` borrows the concept of module from popular DL frameworks such as PyTorch (Paszke et al., 2019) to enable encapsulation of learnable parameters. Conceptually, a module is a function parameterized by learnable parameters that are updated iteratively, for example, by gradient descent. Consistent with the DL practice, `kokoyi-lang` separates learnable parameters from parameters that the function can be applied to using a semicolon: $f(x; \theta)$. Similarly, if $f$ uses a component module $g$ that itself contains learnable parameters, it is written as $f(x; g)$. The syntax of module definition resembles those of LaTeX packages for pseudocode.

Fig. 1(b) demonstrates the major concepts of `kokoyi-lang` with a $L$-layer RNN, where the final hidden representation $h_{[L]}$ embeds the input sentence $S$ that contains the indices of embedding of the tokens. $W_x, W_h, b, h_0$ and the embedding table $D$ are all trainable parameters. Note that $|S|$ is a syntactic sugar to retrieve the first dimension of a multi-dimensional tensor, whereas the more general *GetShape* function returns the complete shape. $\{0\}^1 || S$ combines a tensor construction (of a zero input) and concatenation to right-shift the input. The rest of the codes are all self-explanatory.

$$
\begin{aligned}
e := \; & v & \textit{(variable)} \\
| \; & 0, -2, 1.0, \ldots & \textit{(constant)} \\
| \; & \textbf{let } v \;=\; e \textbf{ in } e' & \textit{(let binding)} \\
| \; & \textbf{letrec } v_1 \;=\; e_1 \,, \ldots, \textbf{in } e' & \textit{(letrec binding)} \\
| \; & \lambda\,(v_1, \ldots, v_k).\, e & \textit{(function abstraction)} \\
| \; & v(v_1, \ldots, v_k) & \textit{(function application)} \\
| \; & \textbf{if } e \textbf{ then } e_1 \textbf{ else } e_2 & \textit{(branch)} \\
| \; & \lambda\,[v_1 \le v \le v_2].\, e & \textit{(functional array)} \\
| \; & \textbf{vmap } v \textbf{ on } v' & \textit{(vectorized map)}.
\end{aligned}
$$

Figure 2: Grammar of KOKOYI IR. $e$ and $v$ represents expression and variable, respectively.

## 4 THE KOKOYI COMPILER

**Intermediate representation specialized for deep learning**  Deep learning compilers have been a popular research topic in the recent years which leaves several choices of Intermediate Representation (IR) for our compiler. Among them, Relay (Roesch et al., 2018) uses functional IR due to its power in expressing function closure; TorchScript (PyTorch, 2021) uses Static Single Assignment (SSA) for its simplicity; JAX (Bradbury et al., 2018) relies on XLA which translates a user program first into High Level Optimizer (HLO) IR for deep learning specific optimizations and then lowers it to LLVM IR.

The KOKOYI compiler chooses functional IR because the mathematical language is inherently very functional, e.g., functions are pure with no side-effect, there is almost no symbol redefinition, etc. This makes it easy for KOKOYI compiler to translate a program into a functional IR by traversing the abstract syntax tree and applying syntactic rules. The full grammar of KOKOYI IR is in Figure 2. Generally speaking, KOKOYI IR follows the A-normal form with two additional rules to capture the common loop patterns in deep learning programs, namely the *vectorized map* (as in Transformer Eq.(1)) and *functional array* (as in RNN Eq.(2)). Having these patterns in IR makes it very convenient to perform optimizations on them later. They demonstrate the necessity of lifting the program interface to a math-like language. While some tools have identified the importance of these patterns and have provided specialized Python APIs for them (e.g., jax.vmap, jax.lax.scan), they typically require complicated specifications from users due to Python's notorious versatility. By contrast, expressing them in math equations is almost out of natural instinct which KOKOYI utilizes to achieve both high usability and efficiency. The translated IR of the attention module is shown in Figure 3(c).

**Strong type or weak type?**  The choice of type system has a significant impact on a programming language and its compiler. The debate on whether a language should embrace a stronger or weaker type system prolongs into the era of deep learning. On one hand, Python's dynamically-typed design saves users from annotating types of variables and functions, which boosts the productivity of model development. On the other hand, lacking static type checking leads to elusive bugs, causing issues like unrunnable codes, buggy codes and reproducibility. Furthermore, deep learning compilers continuously push for stronger type semantics to achieve high performance. For example, TorchScript's script mode requires users to provide type hints for the argument and return values of every method. For more complicated programs, it also provides a trace mode that executes the program once and records the data flow as well as the runtime type information for further optimizations. Program tracing cannot deal with control flows that may diverge during runtime. To circumvent this, AutoGraph (Moldovan et al., 2018) (which is integrated into tensorflow.function) adopts source code transformation to convert control flows to data flow operators. Nevertheless, the transformation rules are limited and can fail in certain cases.

KOKOYI's principle is *usability* oriented. As such, no type hint is required by default. KOKOYI tries to infer the expression type from intrinsic language semantics. Some operators (e.g., matmul) have constraints on the types of inputs and outputs they can be applied to (e.g., tensors). Similarly, the two new IR expressions introduced by KOKOYI, functional array and vectorized map, have list and tensor type respectively. We can propagate these type constraints among the IR nodes to refine

$$\text{Attn}(\{q\}^N, \{k\}^M, d) \leftarrow$$

$$\left\{ \left\{ \text{softmax}\left( \frac{q_{[i]}^T \cdot k_{[j]}}{\sqrt{d}} \right) \right\}_{j=0}^{N} \right\}_{i=0}^{M}$$

(a)

```
1   def Attn(q, k, d):              batch scope: ()
2       M = kokoyi.length(q)
3       N = kokoyi.length(k)
4       _1 = kokoyi.arange(0, M)
5       def _2(i):                  batch scope: (M)
6           _3 = kokoyi.arange(0, N)
7           def _4(j):              batch scope: (M, N)
8               _5 = q[i]
9               _6 = kokoyi.transpose(_5)
10              _7 = k[j]
11              _8 = kokoyi.matmul(_6, _7)
12              _9 = kokoyi.sqrt(d)
13              _10 = _8 / _9
14              return _10
15          _11 = kokoyi.vmap(_4, _3)
16          return kokoyi.softmax(_11)
17      return kokoyi.vmap(_2, _1)
```

(b)

```
letrec Attn = λ q, k, d.
  let M = %%length(q) in
    let N = %%length(k) in
      let a =
        let %1 = %%arange(0, M) in
          letrec %2 = λ i.
            let %3 = %%arange(0, N) in
              letrec %4 = λ j.
                let %5 = %%index_get(q, i) in
                  let %6 = %%transpose(%5) in
                    let %7 = %%index_get(k, j) in
                      let %8 = %%matmul(%6, %7) in
                        let %9 = %%sqrt(d) in
                          let %10 = %%div(%8, %9)
                          in %10
                  in
              let %11 = vmap %4 on %3 in
              let %12 = %%softmax(%11)
              in %12
            in
          let %13 = vmap %2 on %1
          in %13
        in a
  in Attn
```

(c)

Figure 3: (a) Attention module in `kokoyi-lang`; (b) Python code generated by KOKOYI; (c) Intermediate representation of the program.

types that are unknown. Also note that it is not uncommon to see type notations when describing a model during paper writing. Examples are $x : \mathbb{R}^{d_1 \times d_2}$ to specify a matrix-type variable and $f : (\mathbb{R}^{d_1}, \mathbb{R}^{d_2}) \mapsto \mathbb{R}^{d_1 \times d_2}$ to specify a function. It is straightforward for KOKOYI to parse them into extra type constraints in the future, which will also help to auto-generate more initialization codes.

The viable types in KOKOYI are scalar (i.e., integer or floating number), list, tensor, function and tuple. To support type polymorphism (e.g., $x + y$ can be between two scalars, or two tensors, or one scalar and one tensor), KOKOYI iteratively refines expression types by incorporating more precise type constraints. Specifically, the first pass only collects the most general types of each operator, type hints from user annotations as well as from IR requirements. Once the constraints are resolved, we refine the type of each operator according to their signatures, which will introduce more type constraints for further refinement. Type constraints are collected using the *let-polymorphism* (Milner, 1978) algorithm and are resolved by unification. If an expression type cannot be determined statically, KOKOYI will generate code with the most general type and defer the type dispatch to runtime. Otherwise, KOKOYI will generate more precise implementation. For instance, although both RNN (below left) and ReLU (below right) uses the case expression to represent branching, KOKOYI can generate `if...else...` statement for RNN while the vectorized `torch.where` operator for ReLU in most cases.

$$h_{[1 \leq t \leq L]} \leftarrow \begin{cases} h_0 & t = 0 \\ o_{[t]} \circ \tanh(c_{[t]}) & otherwise \end{cases} \qquad \text{ReLU}(x) \leftarrow \begin{cases} x & x \leq 0 \\ 0 & otherwise \end{cases} \qquad (3)$$

## 5 THE AUTO-BATCHING RUNTIME

KOKOYI promotes the concept of *"think in one sample, run in multiple"*. Users express a model as a parameterized function (or a module) over one input sample, from which KOKOYI automatically converts to an implementation that executes multiple samples in a batch. The need for auto-batching not only appears at the scope of an entire program but also at the level of one statement or one expression, or even multiple places that are nested. The code to compute attention in Eq. (1) shows such an example. It calculates attentions between all positions of array $q$ and $k$ by nesting two array definitions and one will wish to compute them in one batch. Moreover, the inner tensor expression can have dynamic length as shown in the decoder attention example.

Another design consideration is debuggability. The best way to avoid bugs is to not having them in the first place, expressing a model in equations already helps to some extent, as we experienced ourselves in prototyping models in KOKOYI. Nevertheless, we cannot realistically hope no bugs creep into the implementation. The success of PyTorch has proven that the *run-and-see* experience is favored by a large population of users, which we wish to keep in KOKOYI. Unfortunately, none of the previous auto-batching solutions can satisfy all the requirements.

In KOKOYI, we build a runtime system that supports nested, dynamic and eagerly executed auto-batching on the top of tensor-based deep learning framework (e.g., PyTorch). Our design is centered around an operational semantics called *General Universal Function (GUFunc)* which specifies how a function over *one* sample can be generalized to multi-sample inputs.

**Operational semantics** GUFunc[1] is broadly adopted by NumPy to generalize functions originally over scalar elements to sub-arrays. A GUFunc specifies the element shape, called *core dimensions*, of its arguments and return values. For example, matrix multiplication is annotated as `(a, b)`, `(b, c) -> (a, c)` meaning that the element shape of the two arguments and the return value is matrix. To support batch processing, a GUFunc allows prepending arbitrary number of *batch dimensions* to its arguments as long as the dimensions satisfy the broadcasting semantics [2]. For example, NumPy's `matmul` supports inputs of shapes $(32, 1, \underline{10}, \underline{5})$ and $(4, \underline{5}, \underline{6})$ respectively because the element shapes (underlined) are valid for matrix multiplication and the batch dimensions $(32, 1)$ and $(4)$ are broadcastable. Conceptually, one kernel conducts $32 \times 4 = 128$ multiplications.

Hence, the auto-batching problem becomes how to make the entire program a GUFunc where its core dimensions specify the shape of an individual sample. To achieve that, KOKOYI first needs to know the number of batch and core dimensions. At the moment, we ask users to provide such information when invoking a KOKOYI-generated function while leaving automatic inference as future work. KOKOYI then ensures that each invoked operator satisfies the GUFunc property with regard to the batch and core dimensionality passed on during execution. For most operators, this is straight-forward as most PyTorch operators have supported broadcasting. For operators working along a certain axis such as reduction or normalization (e.g., `\softmax`), KOKOYI adjusts the working axis according to the batch and core dimensionality. Since the GUFunc property is composable (a function consisting of only GUFuncs is a GUFunc), the entire program can execute all samples in one batch.

**Eager-mode vectorized map** KOKOYI's `vmap(f, x)` is a high-order function with its first argument being a function and the second one being a tensor. Conceptually, it applies `f` over each element of `x` and returns the transformed results in a batched tensor. We call `f` the *mapped function* and `x` the *mapped domain*. The mapped function acts as a transformation routine over one element. As is shown in Figure 3(b), the mapped function `_2` (Line 5-16) computes for one position `i` the attention against all positions `j`. The vmap call (Line 17) applies it over domain variable `_3` which is calculated by `arange(0,M)` (Line 4). This function further contains another nested function `_4` that is mapped by the vmap at Line 15, over the mapped domain `arange(0, N)`.

During execution, KOKOYI maintains a global *batching scope*. Initially, the batching scope has shape `()`, meaning no batch dimensions. `vmap` lifts the current batching scope by appending the batch size of the mapped domain. Whenever a tensor created outside of the current batching scope is referenced inside a mapped function, its batching dimension is expanded by appending the batch size of the current scope. For instance, in Figure 3(a), scalar variable `N` is created at Line 3 outside of any mapped functions. When it is referenced at Line 6 by `kokoyi.arange`, it is automatically expanded to a vector of shape `(M,)`. Because all KOKOYI operators are GUFuncs, they automatically work on a batch of inputs so the computation is vectorized. One subtlety is that the `vmap` itself must be a GUFunc too to allow nested invocation. This is guaranteed by appending the batch size so that the original batch dimensions are retained.

To support dynamically shaped batching as in the decoder attention module, KOKOYI associates each tensor variable with a mask tensor, a boolean tensor indicating whether the corresponding entry is valid. Masks are created by operators that may lead to dynamic shapes within a batch (e.g., `kokoyi.arange`). KOKOYI operators will utilize the mask tensor during computation, e.g., skipping invalid entries during reduction, calculating the result mask or simply dropping it.

---

[1] https://numpy.org/doc/stable/reference/c-api/generalized-ufuncs.html
[2] https://numpy.org/doc/stable/user/basics.broadcasting.html

Naive runtime auto-batching can lead to several performance issues. First of all, expanding batch dimensions may duplicate tensor data. To give an example, suppose matrix `q` (Figure 3(a), Line 1) has shape `(M, D)`, referencing it inside the nested mapped function (Line 8) expands it to be `(M, N, M, D)` where each `(M, D)` sub-tensors are the same. Our implementation resolves the problem by taking advantage of the broadcasting semantics. We unsqueeze `q`'s dimension to `(1, 1, M, D)` and let broadcasting take affect during subsequent operations. Another issue is with the index lookup `q[i]` (Line 8). Although the lookup is vectorized for every element in `i`, the particular case has a more optimized solution using tensor slicing `q[0:M]`. We hereby implement a compiler optimization pass that makes such transformation automatically when detecting the index variable `i` being a range domain.

# 6 EVALUATION

**Diversity and Coverage**  To measure the flexibility of `kokoyi-lang`, we have implemented a variety of popular models, including Multi-layer Perceptron (MLP), LeNet (LeCun et al., 1998), UNet (Ronneberger et al., 2015) for images segmentation, sequence models for sentimental classification, neural machine translation (with and without attention (Sutskever et al., 2014)) using LSTM (Hochreiter & Schmidhuber, 1997), Bi-LSTM (Huang et al., 2015), and finally Transformer (Vaswani et al., 2017). For generative model, we implemented Generative Adversarial Networks (GAN (Goodfellow et al., 2014)) and Variational Auto-Encoder (VAE (Kingma & Welling, 2013)). Finally, in the context of reinforcement learning, we implemented Policy Gradient (PG) and Deep Q-learing Networks (DQN). We have included HTML files of these notebooks in the supplementary.

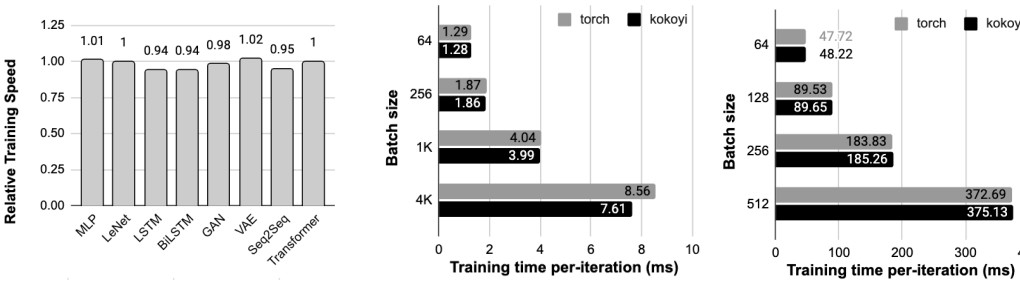

Figure 4: Relative training speed v.s. PyTorch

Figure 5: Per-iteration training time under different batch sizes for MLP (left) and Transformer (right).

**Performance**  We tested KOKOYI's system efficiency on a variety of deep neural network models and compared them with the state-of-the-art well-optimized implementations in native PyTorch (see Appendix C for the model details). We measured the running time of training these models for one iteration, including a forward and a backward propagation pass and the parameter update, and reported the average time of 20 runs. We observed little variance. All the experiments are conducted on an AWS EC2 g4dn.2xlarge instance equipped with a 16GB NVIDIA T4 GPU card. The software used are PyTorch 1.10.0 and CUDA 11.0.

The training speed of these models generally match their PyTorch versions ($\leq 6\%$) as shown in Figure 4, and is stable with larger batch sizes (as in Figure 5), showing the effectiveness of KOKOYI's auto-batching.

**Usability**  Coding in math equations is concise and succinct, especially when the model is complicated. For example, Transformer in `kokoyi-lang` has only 62 lines of code, as compared with 155 in the PyTorch implementation (we count only the forward functions). The key that KOKOYI codes are much compact has to do with the fact that it specifies the computation at a finer granularity and lets auto-batching handles batching, whereas the PyTorch implementation must make sure all operators are batched and aligned, see Appendix-B for a detailed inspection.

| Model | LoC (KOKOYI) | LoC (PyTorch) |
|---|---|---|
| MLP | 10 | 10 |
| LSTM | 25 | 22 |
| Seq2Seq | 30 | 34 |
| Transformer | 62 | 155 |

Table 1: Comparison on line-of-code (LoC) between KOKOYI and PyTorch

## 7 RELATED WORK

**Compilers for deep learning frameworks** are generally divided into two categories: Tensor compilers such as Halide (Ragan-Kelley et al., 2013), TVM (Chen et al., 2018) and Tensor Comprehension (Vasilache et al., 2018) aims at generating efficient codes for tensor operation; Program translators like AutoGraph (Moldovan et al., 2018) and JAX (Bradbury et al., 2018) converts Python into low-level IRs for more efficient execution. KOKOYI can leverage both of them: speeding up individual tensor operator by plugging in functions produced by the tensor compilers; replacing the code generation stage and the runtime with the program translator's for faster overall execution.

**Auto-batching** is a technique that transforms the computation for a single instance to the one for a batch. TensorFlow-Fold (Looks et al., 2017) provides domain-specific languages for users to express sample-wise computation and performs batching on the computation graph, but has limited support for control flow. DyNet (Neubig et al., 2017) generates and batches computation graphs dynamically, but with significant runtime overhead. DGL (Wang et al., 2019) utilizes domain knowledge from the graph neural network family for auto-batching, but cannot generalize to arbitrary programs. (Radul et al., 2019) proposed a local static auto-batching solution that could handle control-flow intensive programs. None of these solutions can handle dynamic shapes.

## 8 FUTURE WORK

There are many possible improvements to the current KOKOYI prototype, including:

**A better `kokoyi-lang` editor.** Users of `kokoyi-lang` may frequently write mathematical notations that are compact after rendering but cumbersome to LaTeX, which necessitates a better editor. There have been many efforts in developing *What You See Is What You Mean (WYSIWYM)* editors for mathematical equations and even LaTeX (Kastrup, 2002), which we believe are good bases to start with. Early users have also expressed making KOKOYI available in other IDEs.

**Effective debugging tools.** Another question is how to present debugging information. Compilers for traditional programming languages associate error messages with the locations to source code. As for KOKOYI, the error messages must be further rendered alongside the highly compact symbol definitions, since the generated backend code has less readability.

**More compiler optimizations.** Many traditional code optimizations for the functional language family are applicable to `kokoyi-lang`, including control-flow analysis, lambda lifting, inlining and so on. Recent advancements such as kernel fusion (Chen et al., 2018) and graph substitution (Jia et al., 2019) that optimize the computation graph of deep learning programs are also promising to integrate.

**Towards a web of executable papers.** Our ultimate vision is to build an ecosystem where papers are *executable*. This can be accomplished by supporting compilation of KOKOYI code snippets inside LaTeX documents. From that perspective, a research paper is nothing more than an elaborated documentation explaining an executable model. Since these snippets are executable, they are guaranteed to be syntactically correct, removing the error-prone process to reproduce their correct implementations. In fact, snippets of KOKOYI codes in different research papers can be cross-linked, making them reusable for the wide community to consume.

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

## A    ESTIMATION OF KOKOYI'S APPLICABILITY

KOKOYI as it stands can mend the gap between math-in-codes and math-in-paper and help researchers who focus on new model development. Of course, not all research fall into such category. We sampled papers from five top machine learning conferences and classified them into the following categories:

1. *Model-NN*: Papers that proposed new model architectures, new loss functions or new neural network building blocks.
2. *Algorithm*: Papers that proposed new algorithm/models but they generally are not implemented with neural networks.
3. *Optim*: Papers with focus on optimization techniques.
4. *Theory*: Papers about theoretical analysis.
5. *Other*: Papers not in the above categories.

Figure 6 shows the result. KOKOYI can potentially benefit most papers in the *Model-NN* category, ranging from 15% to 66%.

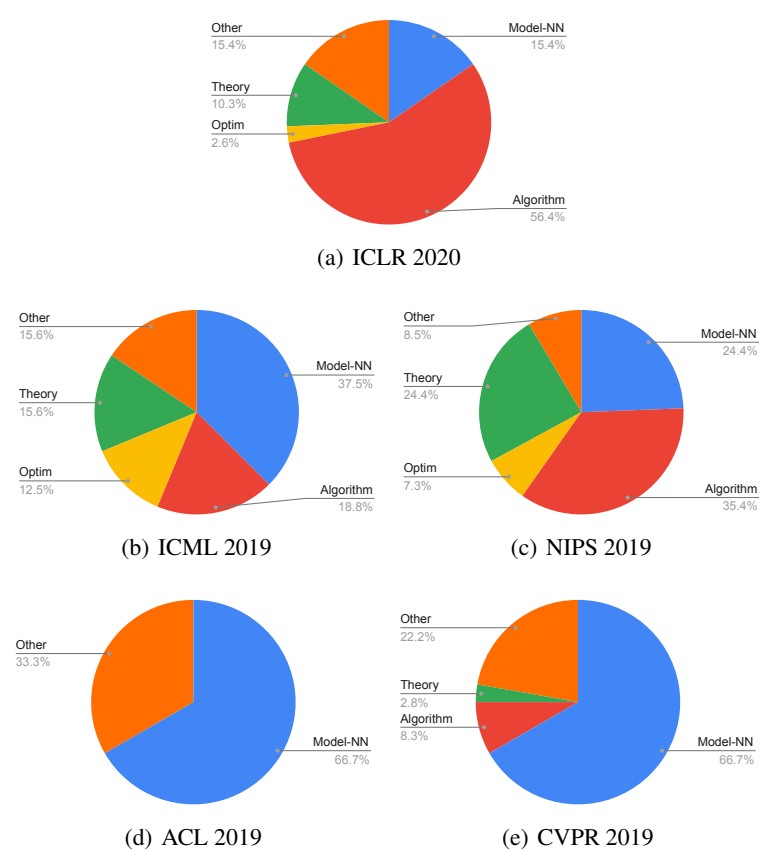

Figure 6: Classification of the papers sampled from several recent conferences

# B  STUDY OF TRANSFORMER CODE

To quantify the actual Line-of-code saving of KOKOYI on the Transformer model, we compare the implementation in PyTorch and KOKOYI side-by-side in Figure 7. Overall, a large portion of the PyTorch implementation is for manipulating shapes of the tensors to align with the batching requirement of the subsequent operations. By contrast, these non-math parts are completely absent in the KOKOYI implementation thanks to the flexible and succinct tensor constructor syntax. Yet, KOKOYI can still achieve similar training speed by its auto-batching design.

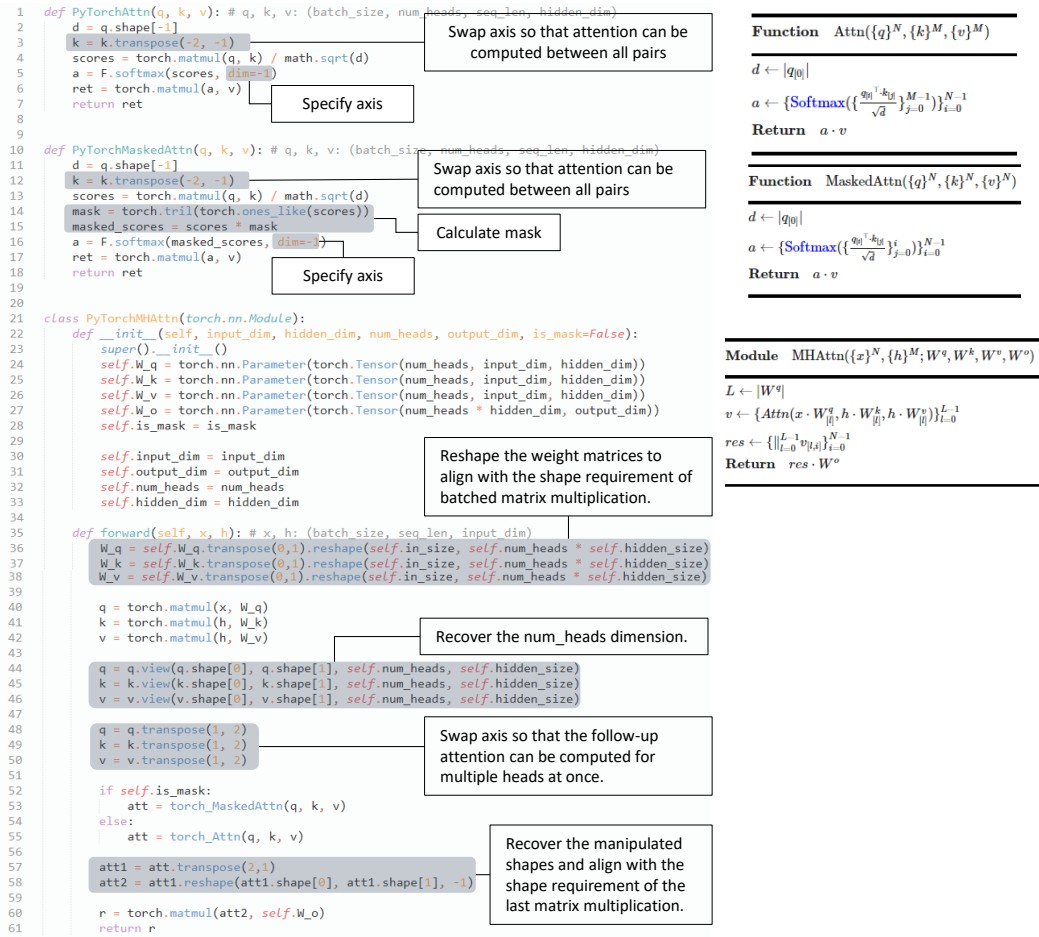

Figure 7: Comparison of the multi-head attention code in PyTorch and KOKOYI. We hightlight the parts of shape manipulation that are saved by KOKOYI.

## C    MODEL BENCHMARK DETAILS

Synthetic inputs are used for measuring training speed. For all the models, we have verified their accuracy on real datasets (see the supplementary). Below lists the model configurations in Figure 4.

- For MLP, we use an input size of 784, a batch size of 512 and all the hidden sizes are also 512.
- For LeNet, we use an input image size of $32 \times 32$ with one channel, a batch size of 64 and all the hidden feature maps have 512 channels.
- For LSTM and BiLSTM, we use an input sequence of length 50, a word vocabulary of size 5000 and a batch size of 512. Both the word embedding size and hidden sizes are set to 512.
- For GAN and VAE, we set the size of the latent vector to be 512 and test them on input images of size $32 \times 32$ with one channel. The batch size is 1024.
- For Seq2Seq and Transformer, both of their source and target sequences have 50 words drawn from a vocabulary of size 5000. Both the batch size and hidden sizes are set to 512.

