# OpenReview forum: "Kokoyi: Executable LaTeX for End-to-end Deep Learning"
_ICLR.cc/2022/Conference — ICLR 2022 Submitted_

### Official Review · Reviewer_cySy · 2021-10-25

**Correctness:** 4
**Technical Novelty And Significance:** 4
**Empirical Novelty And Significance:** 4
**Recommendation:** 6
**Confidence:** 4

**Main Review:**

The paper is well written and clear. It hides many aspects of translation and focuses more on the user side. The idea is extremely fascinating, especially about  "executable papers" which would be very interesting.
Unfortunately, the approach is not testable. I would have appreciated the possibility to test the system to understand how much the language helps to implement my network.
If I have to find a second weakness, probably it is that the paper does not provide formal definitions or details about the implementation. But the message the authors wanted to give is different, thus I cannot consider this as a malus.
Honestly, as said above, I do not see any problem with the paper itself. It is discursive, it tells a story, which in my opinion is a good story, worth discussing. However, it is a story that I have to trust but that cannot play with, which limits the quality of the article.



**Summary Of The Paper:**

The paper presents a language similar to LaTex with a compiler able to translate mathematical formulas of network layers into executable source code. The article considered the PyTorch framework and presented several usability and performance studies.

**Summary Of The Review:**

The paper is good. However, it presents a system that cannot be tested.

---

> ### Author Response · Authors · 2021-11-23
> **Author's response to review #3**
>
> We thank the reviewer for recognizing the novelty of this work. The supplementary contains the HTML files of a major set of Jupyter Notebooks and we will open source the repository soon. We are actively building an interactive web server where one can prototype a model in Kokoyi and download both the generated PyTorch module and the LaTeX snippet. We will post an annonymous link once that is done.

---

### Official Review · Reviewer_8Mk2 · 2021-10-27

**Correctness:** 4
**Technical Novelty And Significance:** 3
**Empirical Novelty And Significance:** 2
**Recommendation:** 5
**Confidence:** 3

**Main Review:**

It is interesting to see a tool that can translate Latex-based math equations/algorithms into Python code. Such a tool could reduce the gap between developing models in math language and implementing them in programming languages. The idea is interesting and the proposed tool is implemented and integrated with Jupyter Notebook.

It seems that the proposed tool is more useful for deep learning (DL) programs that contain a lot of math equations/computations. The developers still need to write other modules, such as data preparation and result visualization, in Python. For less math-heavy programs, the benefits of the proposed tool are not very clear. It is not clear how many DL programs actually contain many math equations and can be benefited from the proposed tool. Many DL programs can directly reuse existing operators/packages without going to the details of math equations (and they are not written from scratch)?

Furthermore, using the proposed tool, developer still need to write latex code. Although the size of latex code is generally smaller than that of Python code, the efficiency/usability of such a tool still needs to be evaluated. For example, to measure the developer productivity when writing code in Python and in Kokoyi.  Furthermore, like Python code,  Latex code could contain bugs too. The correctness of latex implementation cannot be guaranteed, so there is still a gap between model and its implementation.

The paper is generally well written.  A minor point: the following two references appear twice:
Moshe Looks, Marcello Herreshoff, DeLesley Hutchins, and Peter Norvig. Deep learning with
dynamic computation graphs. arXiv preprint arXiv:1702.02181, 2017a.

Graham Neubig, Yoav Goldberg, and Chris Dyer. On-the-fly operation batching in dynamic computation
graphs. In Advances in Neural Information Processing Systems, pp. 3971–3981, 2017a.


**Summary Of The Paper:**

This paper proposes Kokoyi, which can automatically translate mathematics into Python implementations. The proposed tool consists of kokoyi-lang, a programming language with the syntax of LATEX and the semantics of deep learning mathematics, and kokoyi-lang, a compiler and runtime supporting advanced optimizations. Kokoyi is integrated with Jupyter Notebook. To measure the flexibility of Kokoyi, the authors have implemented a variety of popular DL models and performed evaluation.

**Summary Of The Review:**

Pros:
. The proposed idea is interesting.
. The proposed tool is implemented and integrated with Jupyter Notebook.

Cons:
. The benefits of the proposed tool are unclear.
. The evaluation is limited.

---

> ### Author Response · Authors · 2021-11-23
> **Author's response to review #2**
>
> Kokoyi targets researchers, who will need to describe their model in LaTeX code anyways when writing the paper. Our goal is to generate callable Python objects from the LaTeX-based model specification automatically, leading to a net productivity saving, if we consider the end-to-end lifecycle of a project. The reviewer is correct to assert that not all research domains benefit from Kokoyi uniformly. In our survey (Appendix-B), we have listed our findings. Computer vision researchers, for instance, often use diagrams to connect building blocks, whereas NLP researchers typically write out model specification in equations. When applicable, models written in Kokoyi-lang is succinct (due to the absence of batch operations), self-contained and efficient -- this new version includes optimizations that have made Kokoyi-Transformer perform just as well as native PyTorch implementation (see code in Appendix-A, unchanged), along with other benchmarks.
>
> Barring any compiler bugs, there is no gap between a model (in kokoyi-lang) and its implementation (the output of Kokoyi compiler, currently a PyTorch module), in terms of consistency (as well as performance). That leaves bug sites isolated in one place. Rooting out these bugs still takes effort. We argue the succinct presentation in math can help avoiding a portion of such bugs by inspecting the (now more succinct) program (in math equations), while the rest can use whatever tricks one does for a PyTorch module. The debuggability issue is brought up by Reviewer #1 and we responded there.
>
> Thank you for pointing out the problem of the references; we have fixed it in the new version.

---

### Official Review · Reviewer_aMmp · 2021-11-02

**Correctness:** 3
**Technical Novelty And Significance:** 2
**Empirical Novelty And Significance:** 2
**Recommendation:** 5
**Confidence:** 4

**Main Review:**

**Strengths**

The idea of executable papers is interesting, and supporting auto-batching in model implementation languages is a significant problem. The authors present a nice algorithm for the auto-batching problem that has less overhead.

**Weaknesses**

While I understand the appeal to have a LaTeX-based coding language to enable executable papers, I do not feel that it is easier to code in LaTeX rather than code. For most parts, the LaTeX code looks like a fancy new syntax for the actual code (e.g. summations instead of loops), but with several limitations: (i) the lack of state (which the authors also point out) (ii) lack of abstractions, modularity, objects, etc. and (iii) inability to partially evaluate the model (for debugging, again the authors identify this issue). These issues both make it hard for a user to program in kokoyi and might result in inefficient code. For example, since there is no state in the loops (summation),  it is not possible to merge multiple data-processing steps into one loop.

The authors claim that the main benefit is the ability to support auto-batching. While auto-batching is great, it is unclear why it can not be done inside a language like PyTorch with some special syntax (similar to how NumPy does function vectorization). What exactly does a language like LaTeX provide here that facilitate easy auto-batching?


The evaluation is great, but it is missing several details.
 — Why does figure 4 only report bars for 5 models? How does Bi-LSTM, GANs, VAE, PG, and DQN perform?

—What were these models trained on? What is the size of the model, size of the dataset, etc.?

— How does memory usage look like for the two implementations? Is there extra memory overhead because of the translation?

— How many times were the results in Figures 4 and 5 run? What is the standard deviation?

— Table 1 does not report the numbers for all the models.

— A user study evaluating the ease of use of kokoyi would answer whether it is easy to code in python or LaTeX?


**Summary Of The Paper:**

The paper presents a LaTeX-based language and compiler called kokoyi to write math-based models and compile them to actual code (such as PyTorch). The authors present an approach to support optimizations such as auto-batching during this compilation process which significantly reduces user burden. The authors presented kokoyi implementations of several popular models such MLP, CNNs, LSTMs and transformers and showed that the kokoyi compiler does not introduce much performance drop.

**Summary Of The Review:**

I lean towards rejecting this paper. The authors didn’t make a compelling case for why they need a new LaTeX-based language to enable auto-batching. The evaluation also lacks several details.

---

> ### Author Response · Authors · 2021-11-23
> **Author's response to review #1**
>
> > While I understand the appeal to have a LaTeX-based coding language to enable executable papers, I do not feel that it is easier to code in LaTeX rather than code. For most parts, the LaTeX code looks like a fancy new syntax for the actual code (e.g. summations instead of loops), but with several limitations: (i) the lack of state (which the authors also point out) (ii) lack of abstractions, modularity, objects, etc. and (iii) inability to partially evaluate the model (for debugging, again the authors identify this issue). These issues both make it hard for a user to program in kokoyi and might result in inefficient code. For example, since there is no state in the loops (summation), it is not possible to merge multiple data-processing steps into one loop.
>
> In the new version, we have tried to make our goal more clear: “Kokoyi aims at boosting productivity at a community level: researchers code a model *once* instead of twice (one in a program and another in the publication). Kokoyi-lang is designed to be framework agnostic; the model written in kokoyi-lang in a paper is executable by all and can be ported to different frameworks.” That is to say, once a model is written out in kokoyi-lang, it is immediately LaTeX-renderable as well executable as a Python callable object, so productivity-wise it is a net saving. Note that syntaxes such as summation are *not* new, they are how aggregation in math equations are written. Modules and functions are also familiar abstractions.
>
> Kokoyi improves productivity without surrendering performance. In this version, we include new performance data of Transformer that fully matches native PyTorch implementation. Appendix-A (unchanged) shows the Kokoyi version: it is not only shorter but also more succinct, because the programming is per-sample, removing the tedious syntax needed to support batch training.
>
> The old version may not have made it clear what we meant by “state” and “stages”. In this new version, we tried to make it more clear: “compiled kokoyi-lang programs are callable Python objects and can be integrated with other stages (e.g. data preparation, model tuning, and model serving) of the deep learning pipeline”. That is to say, Kokoyi focuses on model definition where stateless computation is almost required due to the need of auto-differentiation.
>
> Since kokoyi-lang programs are compiled into Python objects (and presently PyTorch modules), all debugging tricks in PyTorch still apply. We concede that after compilation readability can suffer, and we can improve it with better naming convention and other needed utilities.
>
> > The authors claim that the main benefit is the ability to support auto-batching. While auto-batching is great, it is unclear why it can not be done inside a language like PyTorch with some special syntax (similar to how NumPy does function vectorization). What exactly does a language like LaTeX provide here that facilitate easy auto-batching?
>
> The point is *not* to provide any extra support and yet achieve the batching capability. Adding new syntax to the existing Python ecosystem inevitably creates a learning barrier while expressing batch computation in math equations like Equation (1) has been widely recognized. Internally, Kokoyi compiler does auto-batching by transforming the user program into PyTorch batchable codes. Thus, a researcher writes the model as if programming against one sample, which is easier to reason about. As our Transformer model shows (Appendix-A), this removes a lot of coding to support batch training.
>
> > The evaluation is great, but it is missing several details. — Why does figure 4 only report bars for 5 models? How does Bi-LSTM, GANs, VAE, PG, and DQN perform? ...
>
> We have included new benchmark results for Bi-LSTM, GAN and VAE. PG and DQN do not use any advanced language features such as vmap and functional array so we omit them due to space constraints. As our data shows, we have matched all performance (including the very challenging Transformer model). There is no extra memory overhead because the generated Python code mimics how a user achieves batch training (e.g., via squeezing/unsqueezing dimensions, etc.). We have added the experiment configurations in Appendix-C.

---

### Author Response · Authors · 2021-11-30
**Live demo of Kokoyi**

Dear reviewers,

Besides the Jupyter Notebook server for an end-to-end experience from code compilation to execution (links only visible to reviewers), we have also prepared a live demo of Kokoyi here: http://35.82.30.143/#/kokoyi . It supports live rendering of your Kokoyi program and compilation into Python code. Feel free to try it out.

---

### Decision · Program_Chairs · 2022-01-20

**Decision:**

Reject

**Comment:**

This paper proposes that ML models might be better expressed in a way closer to their mathematical representation than to Python code.  This is an attractive proposition, but the paper's development of this proposition is that models might best be expressed in LaTeX, which is not a hypothesis that the reviewers consider proven.

Ultimately, this paper proposes a new language in which to express ML models, and compares that language against one baseline: PyTorch.  However, this is far from a reasonable baseline.  Even within Python, systems such as JAX (which the paper dismisses as a "Program translator") are much closer to the pure functional style; and going further afield, comparisons should be to DEX and Julia, to name but two.

The reviewers appreciate the approach to autobroadcasting, but again note that this does not require a new language, and again, systems like JAX, DEX, Julia all have approaches to broadcasting which are not compared.

Reviewer 8MK2 is concerned that we "still need to write other modules .. in Python", but the authors rebut this well: Kokoyi is not expected to be applied to an entire program, just to the model components.

Even if Kokoyi were to be successful, there is a question of its wider applicability.  A major strength of PyTorch/JAX is that they are used by a much larger community than just ML paper authors.  It is because the authors write in these tools that their work is usable by other practitioners.  The paper explicitly says it is targeted not even at ML paper authors, but at a subset of that community.

The usability analyses are very much lacking.  Lines of code is a notoriously coarse tool to assess programming paradigms.  I would also caution against trying to do any small-group user study - the best initial study is to release Kokoyi into the wild, get feedback from users, and then if it proves popular, prepare a paper or monograph.  This is the path of PyTorch and other frameworks.

Until then, the paper may be of interest to a workshop very focused on programming models for ML, but is not currently suitable for the wider ICLR audience.